# BRAINballs Program Improves the Gross Motor Skills of Primary School Pupils in Vietnam

**DOI:** 10.3390/ijerph18031290

**Published:** 2021-02-01

**Authors:** Van Han Pham, Sara Wawrzyniak, Ireneusz Cichy, Michał Bronikowski, Andrzej Rokita

**Affiliations:** 1Department of Team Sports Games, University School of Physical Education in Wrocław, A Mickiewicz Street 58, 51-684 Wrocław, Poland; sara.wawrzyniak@awf.wroc.pl (S.W.); ireneusz.cichy@awf.wroc.pl (I.C.); andrzej.rokita@awf.wroc.pl (A.R.); 2Department of Physical Education, An Giang University, VNU-Ho Chi Minh, 18 Ung Van Khiem, Long Xuyên 90100, Vietnam; 3Department of Didactics of Physical Activity, Poznan University of Physical Education, Królowej Jadwigi 27/39, 61-871 Poznań, Poland; bronikowski.michal.wilk@gmail.com

**Keywords:** TGMD-2, physical education, fundamental motor skills, educational balls, primary school

## Abstract

The purpose of this study was to evaluate the impact of the BRAINballs program on second graders’ gross motor skills in a primary school in Vietnam. A total of 55 students (23 boys and 32 girls) aged seven years participated in the study. The research used the method of a pedagogical experiment and parallel group technique (experimental and control group) with pre- and post-testing. The study was conducted in the school year 2019/2020. The gross motor skills performance was assessed by the Test of Gross Motor Development—2nd Edition. The BRAINballs program was conducted twice a week and combined physical activity with subject-related contents by means of a set of 100 balls with painted letters, numbers, and signs. The results showed that the experimental and control groups improved their motor skills after one school year (*p* < 0.001). However, the analysis of covariance demonstrated that students from the experimental group, compared to students from the control group, showed significantly better scores in both subtests: locomotor (*p* = 0.0000) and object control skills (*p* = 0.0000). The findings of this study show that the BRAINballs program had a positive effect on children’s motor performances and may help to better understand the development of basic motor skills of seven-year-old students in Vietnam.

## 1. Introduction

Fundamental movement skills (FMS)—skills sometimes also called gross motor skills—are considered to be the basic elements for the more advanced, complex movements essential for adequate participation in many physical and athletic activities [1,2,3,4]. FMS are typically classified into locomotor skills (e.g., running), manipulative or object control skills (e.g., catching and throwing), and stability skills (e.g., balancing) [1,5]. An increasing number of studies evidence the association between FMS competence with better health outcomes in children, and this motor proficiency may play a potential role in promoting positive long-term physical activity and health trajectories across the lifespan [6,7,8]. Children who are competent in basic motor skills more willingly and confidently participate in physical and sport activities, and in addition, it helps in the prevention of diseases related to weight [7]. Meanwhile, the recent evidence suggests that failing to acquire FMS at the appropriate age may increase the risk of a child experiencing long-term physical and mental health problems [6]. FMS deficits may influence a pupil’s ability to participate in physical activity, and low levels of physical activity in childhood are associated with many adverse physical and mental health problems [3,9,10]. A recent systematic review concluded that strong positive associations exist between FMS and educational achievements in reading and mathematics [11]. Studies have also linked low levels of FMS with social and emotional problems, including being withdrawn in social settings, having a poor self-concept, higher stress, and increased anxiety levels [12,13].

Research confirms the relevance of proper motor development at an early school age for the sound development of a child [14]. In later childhood (7–10 years), children are very active and enjoy playing, exploring, and discovering new things; for that reason, basic movement skills (walking, running, jumping, balancing, dodging, avoiding, throwing, catching, and kicking) can be easily learned [15]. However, the mastery of FMS does not come naturally [16]; the development must undergo a sequential practice process. The environmental conditions, including opportunities for practice, encouragement, guidance, and education, seem to play an important role in achieving a proper motor proficiency level [17,18,19]. 

Many researchers have also demonstrated that increasing physical activity in the curriculum contributes to better motor performance [20,21,22]. Fisher et al. found that there was a significant correlation between the percentage of time spent on moderate and vigorous physical activity and the scores of basic motor skills [22]. On the other hand, there are a considerable number of studies suggesting that school-based interventions focusing on motor competence enhance children’s FMS. Participation in early motor intervention programs positively influenced children’s motor skills [23,24,25,26,27,28]. Previous research showed that the application of experimental programs focusing on fun games and exercises in the curriculum significantly improved children’s basic motor skills [19,29,30,31]. In general, children should be given opportunities to practice physical activities as soon as possible, participate in learning in a fun and exciting environment and, together with an age-appropriate movement program compatible with their developmental stage, may develop comprehensively their general and specific motor skills. 

Given the above considerations, in Poland, a method called educational balls BRAINball has been developed and successfully applied in several hundred preschools and primary schools [32,33,34]. Educational balls are included on the official list of didactic aids for use in teaching in primary schools and are recognized and approved by the Ministry of National Education in Poland. Currently, the BRAINball is also introduced and known in several countries, such as Germany, Portugal, Finland, Greece, the United States, Singapore, and Taiwan (China) [32]. BRAINball is an innovative teaching method based on an interdisciplinary model of physical education (PE) [35,36]. This method combines PE and academic learning and relies on the development and improvement of children’s motor and academic performance through movement, play, and having fun [34,37,38,39,40,41]. The researchers found that children enjoy playing, moving, and participating in physical activities with the balls. They modified the traditional balls by adding numbers, letters, and mathematical symbols on their surfaces. The size of the balls is also adapted to the children’s (six- to nine-year-old) body sizes [33]. The BRAINball set includes 100 balls for mini team sports games in five colors (yellow, green, blue, red, and orange) with black letters of the alphabet (uppercase and lowercase letters); numbers (from zero to nine); and mathematical symbols (addition (+), subtraction (−), multiplication (*),division (:), greater than (>), less than (<), parentheses (), and the at sign (@)) [34].

The numbers, letters, signs, and colors of the educational balls allow teachers to integrate PE with a variety of contents, such as language (Polish, English, or Spanish); mathematics; history; geography; biology; etc. [34]. Games and exercises with BRAINballs are based on the natural forms of movement (running, jumping, throwing, catching, etc.), and during PE activities, students can easily gain and improve their basic motor skills and develop physical fitness, as well as academic achievements [39,40]. The previous studies showed that PE integrated with subject-related contents that used educational balls helped to develop various skills [32,39]. Children participating in pedagogical experiments with the BRAINballs significantly improved their language skills (reading and writing) [42], math [32], physical fitness [42,43], hand-eye coordination [44], and time-space orientation [45]. 

Thus, the aim of this study was to investigate whether or not teaching physical education with the use of BRAINballs would significantly improve the gross motor skills of seven-year-old Vietnamese pupils in primary school.

## 2. Materials and Methods

### 2.1. Participants

The research sample was second-grade students at Long Xuyen Global International School, An Giang Province (a province in the Mekong Delta region of Southern Vietnam). A total of 55 students (23 males and 32 females) aged 7 years participated in this study. The study was conducted in the school year 2019–2020. The method was pedagogical experiments conducted in natural conditions using the parallel grouping technique. Participants were divided into two groups: 27 students (11 boys and 16 girls) in the control group and 28 students (12 boys and 16 girls) in the experimental group. The teaching process was conducted in both groups (experimental and control) based on the same curriculum specified by Vietnam’s Ministry of Education and Training. Information about this research was provided to the principals, teachers, parents or guardians, and the children themselves before they voluntarily participated. Before participating, parents or guardians signed a consent form for their children to participate in the study. The study was approved by the University Ethics Committee for Research Involving Human Subjects (2009), and all procedures and manipulations were carried out in accordance with the principles of the Declaration of Helsinki.

The experimental factor was a PE program integrated with the BRAINball games and exercises. In the experimental class, all PE lessons twice a week for 35 min were integrated with the BRAINballs for the period of five months (one school semester), and the PE teachers designed lesson plans for each topic in accordance with the curriculum and school activities.

In the control group, PE was twice a week for 35 min and conducted with the traditional curriculum (without BRAINballs). In both groups (experimental and control), PE classes were conducted by the same PE teacher. The teacher had a 10 years’ experience of teaching physical education at the school and, before the pedagogical experiment, was specially trained how to organize and perform games and exercises with BRAINballs.

### 2.2. Research Tool

Gross motor skills were assessed using the Test of Gross Motor Development–Second Edition (TGMD-2). The TGMD-2 consists of two subtests: locomotor skills (run, gallop, hop, leap, jump, and slide) and object control skills (strike, dribble, catch, kick, throw, and underhand roll). Each skill is evaluated based on the performance criteria. Each subtest includes 24 performance criteria. The participant has to perform the task twice. For each trial, a score of 1 is given if the criterion is performed correctly and a score of 0 if performed incorrectly [46]. 

The level of gross motor skills of students from the experimental and control groups was assessed for the first time (pre-test) in September 2019, and the second examination was due in February 2020, but because of the pandemic restrictions, it was impossible to carry out the research as first planned, and therefore, it was decided to postpone the second assessment for another five months (another school semester) to see the potential long-term effects. Therefore, the post-test took place after one school year in September 2020. The tests were conducted during PE classes in the large playground (outdoors). Before the assessment, the exact performances of the 12 gross motor skills of TGMD-2 were explained to the students in detail and demonstrated. After that, each student began to perform each gross motor skill under the supervision of the tester and teacher. The student had to perform two trials for each of the 12 gross motor skills. All testers observed and scored all participants’ performances to assure measurement consistency. The testers scored each performance criteria for each trial on spot.

### 2.3. Data Analysis

For statistical analysis, Statistica software version 13.1 (Dell, Texas, United States) was used. The main dependent variables were the mean scores and standard deviation (SD) for the locomotor and object control skills obtained from the examination of the students from the control and experimental groups. First, using the Shapiro-Wilk test, we confirmed the normality of the distributions of the locomotor and object control skills (*p* = 0.29 for locomotor and *p* = 0.47 for object control, respectively). Then, for comparisons of the changes in the mean parameters of the performed tests within the experimental and control groups, the Student’s *t*-test for dependent samples was used. Next, an analysis of covariance (ANCOVA) was conducted to determine the statistically significant difference between the experimental and control groups after the pedagogical experiment, where the pre-test was set as the covariate. The statistical significance was set at *p* < 0.05.

## 3. Results

The means and standard deviations for the results of pre- and post-tests by groups are presented in Table 1. The results showed that the level of gross motor skills in the experimental and control groups significantly improved after one school year. Students in both groups achieved significantly better results in locomotor and object control skills (*p* = 0.000 for both). In all trials in both subscales, both the experimental and control groups obtained significantly higher scores than at the beginning of school year (*p* < 0.01 for all trials in both subtests) (Table 1).

A one-way ANCOVA was conducted to compare the effectiveness of the BRAINballs program on the students’ gross motor skills. There were significant differences in the total locomotor skills (F = 18.88, *p* = 0.000) and object control skills scores (F = 20.74, *p* = 0,000) between the groups. The experimental group achieved significantly better results compared to the control group in both subtests (Table 2). 

There was also observed a significant effect of the group on the achievements of run and gallop in the locomotor subtest. Students in the experimental group showed significantly better results compared to the control group in both trials (*p* = 0.001 in run and *p* = 0.001 in gallop). In the object control subscale, in three trials, students in the experimental group also achieved significantly better scores than their peers in the control group (*p* = 0.001 in striking, *p* = 0.022 in kicking, and *p* = 0.005 in rolling a ball) (Table 2).

## 4. Discussion

The main purpose of this study was to evaluate the impact of the BRAINballs program using educational balls in physical education classes on second graders in a Vietnamese primary school. The results showed that both groups (experimental and control) significantly improved their gross motor skills after one school year. This means that the current PE program and the BRAINballs program had a positive impact on the children’s locomotor and object control skills. However, comparing the level of influence between the two programs, the results showed that participation in the BRAINballs program significantly improved the students’ skills compared to students in the control group who participated in the traditional PE program, at least in the range of skills that were examined and compared in the study, specifically in the long term. 

Our findings were similar to the results of previous studies demonstrating that physical activity programs focusing on games and exercises to increase the levels of basic motor skills in children can be effective [19,29,30,31]; however, the possible long-term effects were less-recognized. In our research, it was observed that introducing the BRAINballs for five months had a positive effect on the locomotor and object control skill techniques even another five months after ceasing of the influencing factor, which might indicate a sustainable effect. However, this certainly requires further analyses with well-designed longitudinal research. 

Numerous studies also highlighted that school-based interventions focusing on motor competence enhance children’s FMS, and in addition, interventions concentrating on object control skills are more effective [47]. Our results shows that the experimental group significantly improved both their locomotor and object control skills (*p* = 0.0000 for both) after one school year.

The effectiveness of using the BRAINballs method in PE classes has been already proven in previous studies [32,33,34,37,38,39,40,41,42,43,44,45]. The BRAINballs program integrates physical activity and various subject-related contents, e.g., language and math, during PE classes. Participating in this program allows teachers and students to merge the knowledge learned in the classroom with PE contents and activities. Previous studies demonstrated the significant positive relationships between the participation in PE program with educational balls and children’s academic performances in reading and writing [39,42] and math skills [32]. On the other hand, there is strong evidence suggesting a relationship between students’ motor and cognitive developments and highlighting that, the better children’s motor performances, the better their educational outcomes [11,48,49,50,51,52]. Further research is needed, as in this study, we focused only on children’s motor performances. Thus, it seems reasonable to investigate the relationships between students’ gross motor skills and their academic achievements after implementing the BRAINballs program. 

Promoting FMS is integral to a holistic view of development. Researchers suggest optimizing physical, psychological, and mental health by promoting the development of more physically literate children [6]. O’Brien et al. found that adolescents may have a difficult time in making the successful transition towards more advanced skills within the sports-specific stage. The alarming findings indicate that adolescents aged between 12 and 13 years entering their first year of post-primary PE do not display appropriate motor proficiency [53]. It is known that an early identification of motor skill problems is beneficial, and a systematic evaluation may help in the identification of learning difficulties and disorders that can affect the proper development of children [49,54]. Understanding the importance of FMS and the awareness of irregularities in motor skills may help to prevent later school problems, as well as to prepare and implement intervention programs [49,54,55].

The results of this study, once again, confirmed that the BRAINballs program is an exciting and creative teaching method that promotes the holistic development of children’s skills. It seems reasonable to introduce school-based experimental programs or interventions to improve children’s motor and academic performances, especially in early school education. As our findings indicate, participation in PE using the BRAINballs positively improves children’s motor skills, but it may also develop and improve their cognitive skills [32,33,34,39], but future research is needed.

## 5. Conclusions

The use of the “BRAINballs” educational balls in physical education classes significantly improved the motor performances of seven-year-old students in Vietnam. The motor skills of the experimental group improved significantly compared to the control group participating in traditional PE after one year of study with only five months of the stimulus (introduction of the intervening factor—the BRAINball program). The results of this study provide promising early findings that applying BRAINball to the preschool and primary school curriculum in Vietnam could be a useful solution to help improve mobility and physical literacy in the sound and sustainable development of school children, especially in the early education phase.

## Figures and Tables

**Table 1 ijerph-18-01290-t001:** Mean and standard deviation (SD) of the experimental and control groups in the pre- and post-tests.

Subtests Skills	Experimental Group			Control Group		
	Pre-Test	Post-Test	*p*	Pre-Test	Post-Test	*p*
Locomotor	31.54 ± 2.97	38.07 ± 2.14	0.000	31.96 ± 2.85	36.67 ± 2.67	0.000
Run	4.68 ± 1.25	5.86 ± 0.80	0.000	4.96 ± 0.98	5.33 ± 0.83	0.001
Gallop	5.39 ± 0.96	6.50 ± 0.64	0.000	5.56 ± 0.75	6.00 ± 0.68	0.001
Hop	6.64 ± 1.25	7.39 ± 1.07	0.000	6.59 ± 1.05	7.41 ± 0.69	0.000
Leap	4.07 ± 0.66	4.79 ± 0.79	0.000	3.70 ± 0.78	4.85 ± 0.72	0.000
Jump	4.96 ± 1.10	6.29 ± 0.90	0.000	4.96 ± 1.06	6.15 ± 0.82	0.000
Slide	5.79 ± 0.96	7.25 ± 0.70	0.000	6.19 ± 0.68	6.93 ± 0.68	0.000
Object control	30.21 ± 3.12	37.21 ± 2.97	0.000	30.78 ± 3.08	35.70 ± 3.00	0.000
Strike	5.57 ± 1.26	7.29 ± 1.01	0.000	5.70 ± 0.87	6.56 ± 1.05	0.000
Dribble	4.93 ± 1.82	5.68 ± 1.44	0.001	4.93 ± 1.04	5.74 ± 0.81	0.000
Catch	4.75 ± 1.08	5.14 ± 0.89	0.005	4.48 ± 1.19	5.04 ± 0.90	0.008
Kick	4.39 ± 1.75	6.11 ± 0.50	0.000	4.96 ± 0.85	5.85 ± 0.77	0.000
Throw	5.00 ± 0.98	5.86 ± 0.97	0.000	5.11 ± 1.01	6.00 ± 0.96	0.000
Roll	5.57 ± 1.17	7.14 ± 0.89	0.000	5.59 ± 0.93	6.52 ± 0.85	0.000

**Table 2 ijerph-18-01290-t002:** Analysis of covariance for the locomotor and object control skills by group condition.

Subtests Skills	SS	MS	F	*p*	η_p_^2^
Locomotor	39.09	39.09	18.88	0.000	0.266
Run	6.04	6.04	17.30	0.000	0.250
Gallop	4.27	4.27	12.58	0.001	0.195
Hop	0.02	0.02	0.03	ns	0.001
Leap	0.12	0.12	0.21	ns	0.004
Jump	0.26	0.26	0.53	ns	0.010
Slide	1.84	1.84	3.92	ns	0.070
Object control	52.93	52.93	20.74	0.000	0.285
Strike	8.76	8.76	11.65	0.001	0.183
Dribble	0.06	0.06	0.13	ns	0.003
Catch	0.02	0.02	0.04	ns	0.001
Kick	1.87	1.87	5.57	0.022	0.097
Throw	0.09	0.09	0.14	ns	0.003
Roll	5.49	5.49	8.80	0.005	0.145

Note: ns: lack of statistical differences at a level of *p* ≤ 0.05. SS—Sum of Squares; MS—Mean square; η_p_^2^—partial Eta square.

## Data Availability

Data access will be available at the request by contacting pvhan@agu.edu.vn. Same variables are restricted to preserve the anonymity of the study participants.

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
