# Peer review of "BRAINballs Program Improves the Gross Motor Skills of Primary School Pupils in Vietnam"

_ijerph, 2021, doi:10.3390/ijerph18031290_

Round 1

Reviewer 1 Report

Dear authors.
I have read your article carefully and find it very interesting. However, the editing page needs to be slightly improved.
1. On lines 111-113 she is worshiped differently than in the rest of the text.
2. The graphs in figures 2 and 4 are completely illegible.
3. It is worth writing in the text what the numbers in the tables marked in red mean.
4. In Figures 1 and 4, the values of the digits should be placed next to the distribution, the charts will then be easier to read.

Author Response

Dear Reviewer,
We highly appreciate the time you and the second reviewer have taken in evaluating our manuscript. You had very helpful suggestions for the improvements of our manuscript. Given such a great encouragement, we have attempted to deal with all the critical points and we added all required pieces of information (in blue in the MS). You will find a detailed reply to each of these insightful remarks on the earlier version of our manuscript below.
We hope that our responses are satisfactory and we think that, thanks to all reviewers’ comments, our revised manuscript is greatly improved. We believe it is now ready for publication in the International Journal of Environmental Research and Public Health.
Sincerely,
Han van Pham

We are very pleased to know that you found our research very interesting. We are also much obliged to you for all your comments. Below, you will find answers to all of them.

1) On lines 211-213 she is worshiped differently than in the rest of the text.
Thank you for pointing out this issue. We unintentionally wrote the draft name of BRAINballs and we have changed the text.

2) The graphs in figures 2 and 4 are completely illegible. It is worth writing in the text what the numbers in the tables marked in red mean. In Figures 1 and 4, the values of the digits should be placed next to the distribution, the charts will then be easier to read.
We are grateful for these comments. We agree that the figures were illegible. We have decided to rewrite the Results section and in the revised manuscript we have inserted two tables with the most important information.
As to the quality of the English language, the revised version was checked linguistically by two proficient Polish-English bilinguals who have several of the most important international English language certificates.

Reviewer 2 Report

The introduction is too exhaustive for an article, it should be written more schematically, and avoid obvious things. example line 71: (order 71 number: 1566/2003 - on the basis of ordinance of the Ministry of National Education and 72 Sport - Diary Acts of 2002, No .69, item 635)

Regarding the analysis of the results, I do not agree.
In a longitudinal study, the homogeneity of the groups must be carried out at the beginning, and then an analysis of the covariance of the differential scores should be carried out, taking the pretest as a covariate.

Regarding the presentation of the results, the graphs do not contribute anything, they must be replaced by tables with mean and standard deviation, and the result of p., And a post hoc test when necessary.

The tables presented do not make much sense, and do not explain any results. It would be necessary to recode them with the necessary information that provides knowledge.
So the results derived from them are meaningless.

Once the analysis and arrangement of the results have been modified, the discussion and conclusions must be varied.
Therefore, a new analysis of the data is necessary, and to modify everything that follows.

Author Response

Dear Reviewer,
We highly appreciate the time you and the second reviewer have taken in evaluating our manuscript. You had very helpful suggestions for the improvements of our manuscript. Given such a great encouragement, we have attempted to deal with all the critical points and we added all required pieces of information (in blue in the MS). You will find a detailed reply to each of these insightful remarks on the earlier version of our manuscript below.
We hope that our responses are satisfactory and we think that, thanks to all reviewers’ comments, our revised manuscript is greatly improved. We believe it is now ready for publication in International Journal of Environmental Research and Public Health.
Sincerely,
Han van Pham

We are much obliged to you for all your comments. Below, you will find answers to all of them.

1) The introduction is too exhaustive for an article, it should be written more schematically, and avoid obvious things. example line 71: (order 71 number: 1566/2003 - on the basis of ordinance of the Ministry of National Education and 72 Sport - Diary Acts of 2002, No .69, item 635)
Thank you also for pointing out that. As suggested by you, the Introduction section has been rewritten.

2) Regarding the analysis of the results, I do not agree. In a longitudinal study, the homogeneity of the groups must be carried out at the beginning, and then an analysis of the covariance of the differential scores should be carried out, taking the pretest as a covariate.
As you recommended, we have run the new analysis. We confirmed the normality of the distribution with the use of the Shapiro-Wilk test, and then an analysis of covariance was conducted to determine a significant difference between groups after pedagogical experiment.
3) Regarding the presentation of the results, the graphs do not contribute anything, they must be replaced by tables with mean and standard deviation, and the result of p., And a post hoc test when necessary.
We are grateful for these comments. We agree that the figures were illegible. As you recommended, we have prepared such a table and included it.

4) The tables presented do not make much sense, and do not explain any results. It would be necessary to recode them with the necessary information that provides knowledge. So the results derived from them are meaningless.
Thank you for pointing out this issue. As you suggested, we have inserted a table with summary results of analysis of covariance.

4) Once the analysis and arrangement of the results have been modified, the discussion and conclusions must be varied. Therefore, a new analysis of the data is necessary, and to modify everything that follows.
As you recommended, we have run a new analysis and the Results section has been rewritten. We have also modified the Discussion and Conclusion section and added some information important in this undertaking topic.

Round 2

Reviewer 2 Report

The authors have made the suggested changes, and the article is easier to read and more understandable.

I think it would be important to add this recent reference in one or in the two texts indicated below (lines 72 and 85), since they link cooperative learning and emotional intelligence, which is very important:

https://doi.org/10.3390/ijerph17145090

line 72: "This method combines PE and academic learning, and relies on the development and improvement of children’s motor and academic performance through movement, play and having fun [35, 38-41]."

line 85: "during PE activities students can easily gain and improve basic motor skills and develop physical fitness and as well as academic achievements [40-41].

Author Response

Dear Reviewer,

We are very pleased to know that you are satisfied with all the changes we have made.

Thank you for pointing out this publication. We added it in line 74: “This method combines PE and academic learning, and relies on the development and improvement of children’s motor and academic performance through movement, play and having fun [35, 38-42] (in blue in MS).

We hope that our responses are satisfactory and we believe that our manuscript is now ready for publication in the International Journal of Environmental Research and Public Health.

Sincerely,
Han van Pham